# Real-Time Fire Smoke Detection Method Combining a Self-Attention Mechanism and Radial Multi-Scale Feature Connection

**DOI:** 10.3390/s23063358

**Published:** 2023-03-22

**Authors:** Chuan Jin, Anqi Zheng, Zhaoying Wu, Changqing Tong

**Affiliations:** 1School of Sciences, Hangzhou Dianzi University, Hangzhou 310018, China; 211070104@hdu.edu.cn (A.Z.); tongchangqing@hdu.edu.cn (C.T.); 2Southeast-Monash Joint Graduate School, Southeast University, Suzhou 210096, China; zwuu0079@student.monash.edu

**Keywords:** fire smoke detection, multi-scale feature, attention mechanism, radial connection, cross-grid matching strategy, weighted decay

## Abstract

Fire remains a pressing issue that requires urgent attention. Due to its uncontrollable and unpredictable nature, it can easily trigger chain reactions and increase the difficulty of extinguishing, posing a significant threat to people’s lives and property. The effectiveness of traditional photoelectric- or ionization-based detectors is inhibited when detecting fire smoke due to the variable shape, characteristics, and scale of the detected objects and the small size of the fire source in the early stages. Additionally, the uneven distribution of fire and smoke and the complexity and variety of the surroundings in which they occur contribute to inconspicuous pixel-level-based feature information, making identification difficult. We propose a real-time fire smoke detection algorithm based on multi-scale feature information and an attention mechanism. Firstly, the feature information layers extracted from the network are fused into a radial connection to enhance the semantic and location information of the features. Secondly, to address the challenge of recognizing harsh fire sources, we designed a permutation self-attention mechanism to concentrate on features in channel and spatial directions to gather contextual information as accurately as possible. Thirdly, we constructed a new feature extraction module to increase the detection efficiency of the network while retaining feature information. Finally, we propose a cross-grid sample matching approach and a weighted decay loss function to handle the issue of imbalanced samples. Our model achieves the best detection results compared to standard detection methods using a handcrafted fire smoke detection dataset, with $AP^{val}$ reaching 62.5%, $AP_{S}^{val}$ reaching 58.5%, and FPS reaching 113.6.

## 1. Introduction

In modern society, fire poses significant threats to human life and health, economic development, and environmental protection [1,2]. Early detection of fires is of the utmost importance since the damage caused by fires tends to grow exponentially over time [3]. Smoke often appears before and accompanies a fire, and enhancing the detection of smoke by detectors can effectively prevent the spread of fire. However, the shape, characteristics, and scale of flames and smoke are not constant, and the environment of scenes where fires may occur is exceedingly complex. This greatly inhibits the detection effect of photoelectric or ionization-based detectors. Moreover, these detectors can only detect the presence of fire and smoke, but they cannot provide information about the location and size of the fire source. Furthermore, they are unsuitable for outdoor scenes. As a result, the accurate and timely detection of the generation and location of flames and smoke in natural scenes is crucial to safeguard people’s lives, property, and the social and ecological environment.

The existing methods for fire smoke detection can be broadly categorized into two main approaches: traditional methods that involve the manual design of image feature information and deep learning methods that automatically extract image features. Traditional methods typically rely on underlying image features such as morphology, colour, and texture. For example, a robust accelerated feature that reflects texture characteristics [4], shape context that represents contour shape [5,6,7,8], sparse coding based on visual feature construction [9,10], background contrast object detection dependent on optical flow difference and wavelet variation [11,12,13,14,15,16,17,18,19], and Gabor filters for extracting object edge features for texture analysis [20,21,22]. While manually designed feature-based algorithms can detect fires to a certain extent, they are often hardware- and environment-intensive. They may fail when the lighting changes too quickly, be unable to detect overlapping targets between adjacent frames, lack awareness of dynamic changes in the external background environment, and require significant computational resources, making real-time detection challenging.

Deep learning-based methods for detecting fire smoke have utilized neural networks and loss functions for training on large datasets. With the help of convolutional neural networks (CNNs), features can be automatically extracted, leading to continuous improvements in object detection accuracy. Typically, deep learning algorithms for fire smoke detection divide the input image into multiple regions and determine if those regions contain the target to be detected. They then adjust the region boundary to more accurately predict the true bounding box of the target. Two primary detection methodologies are anchor-based and anchor-free [23]. The former generates multiple candidate anchors of different sizes and proportions at the same pixel point, each responsible for detecting objects whose intersection ratio is greater than a specified threshold. The latter locates the key point or centre of the object directly without clustering multiple anchor templates with different aspect ratios on the current training data before training. Deep learning-based fire smoke detection algorithms, such as those proposed in recent studies [24,25,26,27,28,29,30,31,32], not only improve detection accuracy but also speed up the detection process, making them suitable for real-life applications. Deep learning-based methods improve the accuracy of fire smoke detection by avoiding the limitations caused by manual feature design in traditional methods. Furthermore, they can automatically learn feature information from large datasets, resulting in highly robust detection systems. Although deep learning-based methods have greatly improved object detection accuracy, they heavily rely on a large amount of training data to ensure accurate real-time detection. In real-world scenarios, beyond detecting fire and smoke in typical environments, a significant amount of detection work needs to be conducted in harsh environments such as fog, rain, and those with bright illumination, all of which can increase the false detection rate of the detector.

In this paper, we propose a real-time fire smoke detection algorithm based on multi-scale feature information and an attention mechanism, which can effectively solve its scale variation and the problem of difficult feature extraction. The main contributions of this paper are as follows:We constructed a dataset containing hundreds of real-life scenarios featuring fire smoke. To mitigate the impact of environmental factors, we also included negative samples, such as sun, clouds, smoke, and lighting, among others.We propose improved structures and strategies specifically designed for detecting fire smoke characteristics. Our improved structures include CDPB, RC FPN-PAN, and PSA, which aim to enhance the model’s feature extraction capabilities. In addition, we have developed a cross-grid matching strategy and a weighted decay loss function to address the unbalanced matching of positive and negative samples that often causes problematic convergence of the loss function.We conducted sufficient experiments to verify a number of possibilities that could affect the results of fire smoke detection.

## 2. Related Works

### 2.1. Multi-Scale Feature

Object detection is a challenging task because of the inherent variability in the shapes, features, and scales that need to be detected. Some objects may be extremely small or large or have non-standard shapes. CNNs are often used to construct high-level networks that capture rich semantic information. However, they may have low-resolution feature maps and weak geometric details. On the other hand, low-level networks can have high resolution and abundant geometric information but limited semantic information and small receptive fields. To address the multi-scale detection problem, a common approach is to predict small objects on high-resolution feature maps and large objects on low-resolution feature maps, leveraging high-level semantic information to improve accuracy. Multi-scale features can be generated using image or feature pyramid structures, as illustrated in Figure 1. The key idea is to use these multi-scale features to detect objects of different scales.

#### 2.1.1. Image Pyramid Structure

The image pyramid structure enables an image to be interpreted at multiple resolutions by scaling the image to different resolutions for multi-scale pixel sampling. During feature extraction in a CNN, feature maps of various sizes can focus on distinct regions of the picture because the size of the CNN receptive field is fixed. The receptive field corresponding to the high-resolution feature map is closer to small objects, making high-resolution images more suitable for detecting small objects. Similarly, low-resolution images are more suitable for detecting large objects. Image pyramids can effectively address the problem of scale variation of targets and improve object detection performance. However, the computational overhead associated with image pyramids is substantial. Each input image of different resolutions must pass through the same CNN, resulting in significant redundant computation. Using larger batch sizes for training is challenging, which can impact model accuracy. Furthermore, inference time increases exponentially, raising the threshold for practical algorithm implementation.

#### 2.1.2. Feature Pyramid Structure

While CNNs are effective for many image recognition tasks, they struggle with object-scale transformations due to their translation invariance. Moreover, the rich semantic information captured by high-level networks during multi-layer feature extraction and compression can result in the loss of small-scale details. To address the computational redundancy of image pyramid extraction, T. Lin et al. proposed a feature pyramid using convolution 1 × 1 and summing the results with top-down connections that have been upsampled [33]. The top-down part generates coarse-grained features, and the bottom-up adds fine-grained features through lateral concatenation. By inputting a single-resolution image, feature maps of different resolutions are obtained. Strong semantic information at low resolution and weak semantic information but rich spatial information at high resolution are fused with a small additional computation. The standard methods for obtaining multi-scale feature maps of images are multi-scale feature [34,35], multi-scale feature fusion with single-scale feature prediction [36,37,38], and multi-scale feature fusion with multi-scale feature prediction [33,39,40,41,42,43]. The main idea is to gradually aggregate adjacent layers from deep to shallow and use the multi-scale features generated by this process for predictions.

### 2.2. Attention Mechanisms

The attention mechanism learns a set of weighting coefficients from the network and dynamically applies them to highlight regions of interest while suppressing others. In computer vision, attention mechanisms can be classified into two main categories: hard attention mechanisms and soft attention mechanisms [44]. The hard attention mechanism is a stochastic prediction, where every point in the image has the potential to extend attention and emphasize dynamic changes. However, its application is limited due to its non-differentiable nature. In contrast, the soft attention mechanism is differentiable everywhere and can be obtained through gradient backpropagation training of neural networks, making it more widely used. Soft attention mechanisms can be further categorized into three types based on their dimensions: channel attention, spatial attention, and self-attention, as illustrated in Figure 2.

#### 2.2.1. Channel Attention

Channel attention is a technique that aims to capture relationships between distinct feature maps, learn the importance of each feature channel through network training, and assign different weights to each channel. One popular module for channel attention is the squeeze-and-excitation (SE) module proposed by J. Hu et al., which adaptively re-weighs feature responses across channels [45]. Another module, called the efficient channel attention (ECA), was introduced by Q. Wang et al. [46]. The ECA module uses one-dimensional sparse convolution operations to optimize the fully connected layers used in the SE module. This results in a significant reduction in the number of parameters while maintaining comparable detection performance. In contrast to the SE module, the ECA module simplifies the interaction between channels by allowing each current channel to only interact with its k surrounding channels. The SE module, on the other hand, uses two multi-layer perceptrons to learn the correlation between different channels, forming an intensive connection between feature maps. Both modules aim to compress the number of parameters and improve computational efficiency, with the SE module adopting a dimension reduction strategy first and then a dimension increase.

#### 2.2.2. Spatial Attention

Spatial attention aims to enhance the representation of crucial regions in the feature maps. It achieves this by transforming the spatial information in the original image into another space, generating a weight mask for each position, and weighing the output to enhance specific regions of interest while suppressing other irrelevant background regions. The convolutional block attention module (CBAM) structure, proposed by S. Woo et al., combines channel and spatial attention. It connects a spatial attention module (SAM) to the original channel attention block, allowing for feature aggregation along both dimensions [47]. SAM is based on the global average pooling (GAP) and global maximum pooling (GMP) operations, which provide two feature maps reflecting different information to act back on the original input feature map, thus allowing for the enhancement of the target region. Another method for spatial attention is A${}^{2}$-Nets, proposed by Y. Chen et al., which aggregates essential attributes from the entire space into a compact collection and then adaptively distributes them to each location [48].

#### 2.2.3. Self-Attention

Self-attention is a unique form of attention that aims to reduce dependence on external information and maximize the utilization of inherent feature information for attentional interaction. Typically, self-attention is achieved by mapping the original feature map into three vectors: the query matrix, the key matrix, and the value matrix. Firstly, the relevance weight matrices of the query and key matrices are calculated. Secondly, the weight matrices are normalized. Finally, the weight coefficients are combined with the value matrices to model global contextual information [49]. X. Wang et al. applied self-attention to computer vision and proposed the non-local (NL) module, which captures long-range feature dependencies by modelling global context through a self-attention mechanism [50]. The dual attention (DA) module, proposed by J. Fu et al., integrates both the channel attention module and the spatial attention module, using channel features and spatial pixel points as query conditions for context, respectively, to adaptively combine local features and global dependencies [51]. Y. Cao et al. proposed the global context (GC) module, which combines SE and a simplified spatial attention module to replace the original spatial downsample process and reduce the computational effort of compressing feature maps based on global pixel points using 1 × 1 convolution [52].

## 3. The Proposed Method

### 3.1. Overview

We propose an efficient detection method based on deep learning to address the scale variation of fire and smoke. Considering the real-time nature and high transparency of fire smoke, our method is primarily based on the latest YOLOv7 model structure [53]. It combines an attention mechanism, a re-parameterization convolutional structure, and a simplified ConvNeXt module [54], as shown in Figure 3. The model consists of several key modules: CBS, ELAN, SPPCSCP, PSA, and REP. The CBS module includes a convolutional layer, BN layer, and SiLU activation function. The ELAN module is composed of nine CBS and consists of two branches, one of which passes through six more CBS with a two CBS overlap between each. Finally, the two branches are spliced, and the output size is adjusted through CBS. The SPPCSCP module comprises a CBS and a pooling layer, which extends the perceptual field and allows the model to adapt to images of different resolution. The permutation self-attention mechanism (PSA) aims to obtain contextual information about all the pixels in its cross path. Through further recursive operations, each pixel can finally capture the dependency of the entire image. By incorporating these improvements, we significantly enhance the model’s fire and smoke detection efficiency.

### 3.2. FPN-PAN Structure with Radial Connection

During the initial fire, the visual source was so small that it occupied only a few pixels in the image. The feature fusion network was enhanced by incorporating lower-level feature information through weight adjustments and cross-layer connections to capture more information about this tiny object. Additionally, the network structure parameters were fine-tuned to improve the feature fusion. In the YOLOv7 model, the neck component utilizes an FPN-PAN structure, which adds a bottom-up route to the FPN, while the FPN layer provides top-down semantic features, the PAN conveys bottom-up positional features, enabling the deeper features to leverage location information at the bottom [55]. However, in this structure, all feature information is given equal weight, even though each information contributes differently to the final output. As a result, the model treats the background information of the image with the same attention as the foreground information, which lowers its generalization performance.

To address the challenges mentioned above, we have modified the feature extraction phase of the backbone network. Specifically, we have added the weights of the FPN-PAN to the feature values of the early layers. This modification, illustrated in Figure 4, facilitates the integration of features from three distinct scales into the same scale. By directly leveraging multi-scale information, this approach helps prevent unnecessary information loss. The revised network topology involves two key steps. First, layers P3 to P5 are added to C4 to C6 through an upsample, or top-down summation, after the preceding convolution process. Second, layers P5 to P3 are added to the former through the PAN structure, or bottom-up summation operation, to create the FPN-PAN structure. The outputs of the initial layers C3 to C6 and the FPN are then added layer by layer through shortcut branches to the input of the PAN structure, as shown by the dashed line in Figure 4. Finally, the output feature channels are transformed into corresponding dimensional values using a 1 × 1 convolutional layer.

The P3, P4, P5, and P6 feature layers are generated through deep convolutional procedures, and they decrease in size as the number of channels increases. The low-resolution feature layer typically represents the semantic features of an object, whereas the high-resolution feature layer represents its contour features. As a result, information pertaining to small targets is usually found in the high-resolution feature layer. In this paper, we augment the network neck with the high-resolution P3 feature layer, which increases the number of detection heads in the original model from three to four.

### 3.3. CDPB Structure

Feature extraction networks play a crucial role in improving models’ detection efficiency and accuracy. CNNs have gained widespread attention and carried out many pioneering works in the field of computer vision under their ability to accurately capture high-level semantic information. However, with the continuous development of the Transformer model, which has the advantages of scalable behaviour and multi-headed attention mechanisms, it has gradually replaced CNNs in terms of function and is widely used in computer vision downstream tasks, such as image classification, object detection, and semantic segmentation. However, the use of attention mechanisms leads to an abrupt increase in model complexity. The visual transformer (ViT) model generates a single low-resolution feature map with a computational complexity that is quadratic to the input image size [56]. The shifted windows transformer (SwinT) model, on the other hand, solves this problem using a hierarchical structure similar to that used in CNN and local attention to achieve a computational complexity linearly related to the size of the input image, indirectly demonstrating the importance of CNN in feature extraction [57]. Therefore, the ConvNeXt module is built on a normalized convolutional structure, as shown in Figure 5, borrowing the structure and training model of the Transformer model in order to obtain high accuracy and scalability.

Detection efficiency is crucial, as fire smoke detection is typically utilized in particularly complicated environmental contexts. Therefore, we propose a new feature extraction structure based on the ConvNeXt block using DWConv and PWConv instead of the traditional full convolution module, called the CDPB, as illustrated in Figure 5. The two 1 × 1 convolutional layers of the ConvNeXt module are replaced with a PWConv structure to lower the computational complexity while guaranteeing the model’s accuracy. The structure primarily comprises three 1 × 1 convolutional layers, two PWConv layers, and one DWConv layer. For the input feature, firstly, it passes through two separate convolutional layers, one of which does not change the original channel structure, and the other performs a scaling operation in the depth direction; Secondly, the DWConv structure is applied to the depth-scaled features, which are then passed through the LN layer, as well as rearranging the channel order of the features to obtain the information interactions in different dimensions. Subsequently, the two PWConv structures are applied so that they satisfy the linear bottleneck structure. Finally, the feature matrices are summed by utilizing residual concatenation, which effectively prevents the gradient disappearance and explosion while retaining the original feature information as much as possible.

Furthermore, due to the variation in image sizes in the fire smoke dataset, it is necessary to ensure that the images input into the model are fixed for practical training. However, the original YOLOv7 model utilized multiple pooling operations to achieve this, resulting in a significant loss of semantic information about the objects. This poses a challenge for small target detection problems where the fire source may be visually limited, leading to poor model performance. Early detection of fire sources is crucial in practical scenarios, as fire damage tends to escalate rapidly over time, making it difficult for the model to meet practical application requirements. To address this challenge, the CDPB module replaces the pooling operation before the ELAN module in the original YOLOv7 structure, which comprises one GMP and three CBS. The CDPB module leverages DWConv and PWConv structures, which require fewer parameters and better capture information between different channels, resulting in more accurate and efficient fire source detection.

### 3.4. Permutation Self-Attention

For channel attention, it is easy to ignore information interactions within space as it is a global processing of information within a channel, while spatial attention treats the features in each channel equally, which easily ignores the information interactions between channels. Therefore, in this paper, we consider using self-attention, as shown in Figure 6, to capture the remote contextual information in both horizontal and vertical directions as accurately as possible. Specifically, the input image is passed through a deep convolutional neural network to generate a feature matrix *M*. The reduced feature matrix is first passed through a convolutional layer to obtain a reduced-dimensional feature matrix, and then the reduced feature matrix is replaced by a permutation operation to generate a new feature matrix $M^{{}^{\prime }}$. Finally, the long-distance contextual information is aggregated $M^{{}^{\prime }}$, and each pixel is synchronised in a cross displacement.

For the local features of the model $M\in R^{C\times H\times W}$, PSA first acts on *M* with three 1 × 1 convolutional layers to generate, respectively, the three feature matrices *Q*, *K*, and *V*, where $Q,K\in R^{C/r\times H\times W}$, $V\in R^{C\times H\times W}$, and *r* represent the corresponding scaling factors. After obtaining the feature matrices *Q* and *K*, the attention matrix $M^{{}^{\prime }}\in R^{H\times W\times (H+W)}$ is further generated by the simulation calculation, which is calculated as
(1)$$\begin{array}{c} d_{i,u}=Q_{u}K_{i,u}^{⊤} \end{array}$$
where $Q_{u}$ represents the feature vector corresponding to position *u* in the feature matrix *Q*, $K_{u}$ refers to the set of horizontal and vertical pixels at position *u* in *K*, $K_{i,u}$ refers to the pixel points in the set $K_{u}$ corresponding to position *i*, and $d_{i,u}\in M^{{}^{\prime }}$ represents the correlation between the feature vector $Q_{u}$ and $K_{i,u}$. The softmax function is then applied along the channel dimension on $M^{{}^{\prime }}$ to calculate the resulting attention map.

The information-gathering operation of remote context is performed on the spatial dimension of the obtained attention graph in the feature matrix *V*. Finally, using the idea of residuals together with the input features *M*, a rich feature representation $M^{\prime \prime }$ is obtained, which is calculated as follows:(2)$$\begin{array}{c} {M_{u}}^{\prime \prime }=\sum_{i\in |Z_{u}|}M_{i,u}^{{}^{\prime }}Z_{i,u}+M_{u} \end{array}$$
where $Z_{u}$ is the feature vector corresponding to position *u* in the feature matrix *V*, $Z_{i,u}$ refers to the pixel points in the set $Z_{u}$ corresponding to position *i*, and ${M_{u}}^{\prime \prime }\in R^{C\times H\times W}$ denotes the feature vector of the feature matrix at *u*. After the displacement, an operation obtains the contextual information of all the pixel points on the cross path of the local feature *M*, and then through further recursive operations, each pixel point can eventually capture the full image dependency.

### 3.5. Matching Strategies and Loss Functions

#### 3.5.1. Dynamic Matching Strategies across Grids

To overcome the problem of unbalanced matching of positive and negative sample sizes at the time of the model, this paper uses a dynamic cross-grid matching strategy combining YOLOv5 and YOLOX. The cross-grid matching strategy of YOLOv5 is to find the two nearest target centroids from the top, bottom, left, and right grids of the current grid, together with the current grid for a total of three grids to match, increasing the number of positive samples and speeding up the convergence of the model. On the other hand, SimOTA dynamic matching strategy in YOLOX [58] calculates a cost matrix that represents the cost relationship between each true bounding box and each feature point. The aim is to adaptively find the true bounding box to which the current feature points should be fitted most. The higher the overlap, the more accurate the classification, and the more within a certain radius.

The SimOTA dynamic matching strategy’s first step, which used the centre prior, has been replaced by the YOLOv5 cross-grid matching strategy. The new approach first boxes all template points in the three grids obtained by the YOLOv5 cross-grid matching strategy. A fixed centre region of size 5 × 5 is set within the true bounding box position. The template points within the true bounding box and the fixed centre region are pre-screened targets. Next, the *k* candidate frames with the highest IoU are selected based on the obtained IoU matrix, where *k* is the minimum value of 10 and the current corresponding region template points. Afterwards, the *k* template points with the lowest cost loss values are assigned to each true bounding box. Only the template points within the fixed central region of the current true bounding box correspond to a lower cost loss. The remaining template points must have a large loss because the assigned weights are too large. The lowest template points are selected according to their loss values. An allocation matrix is then constructed, which records the positive sample corresponding to each true bounding box. The position of the corresponding positive sample is marked one, and everything else is marked zero. Finally, the prediction box is compared using the loss value, and the smaller value is selected for further screening to ensure that a true bounding box is allocated to only one template point.

Based on the process mentioned above, positive samples and their corresponding true bounding boxes can be identified, while the rest are classified as negative samples. This dynamic matching strategy across the grid provides more precise prior knowledge of the current model’s performance to screen the predicted bounding boxes, as opposed to the original positive and negative sample classification strategy.

#### 3.5.2. Weighted Decay Loss Function for Object Detection

Object confidence loss and classification loss in YOLOv7 adopt logarithmic binary cross-entropy loss, while coordinate loss uses CIoU loss, which is calculated as follows:(3)$$\begin{array}{c} Loss_{CIoU}=1-CIoU+\frac{\rho^{2}\left(b,b^{gt}\right)}{c^{2}}+av \end{array}$$
where
(4)$$\begin{array}{c} IoU=\frac{A\cap B}{A\cup B} \end{array}$$
(5)$$\begin{array}{c} a=\frac{v}{1-IoU+v} \end{array}$$
(6)$$\begin{array}{c} v=\frac{4}{\pi^{2}}{\left(arctan\frac{w^{gt}}{h^{gt}}-arctan\frac{w}{h}\right)}^{2} \end{array}$$*A* and *B* indicate the coordinate information of the bounding box, $\rho^{2}\left(b,b^{gt}\right)$ denotes the Euclidean distance between the centre point of the prediction frame and the real frame, and *c* denotes the diagonal distance of the smallest closed area that can contain the prediction frame and the real frame.

To address the issue that the loss function is difficult to converge due to the unbalanced division of positive and negative samples, this paper offers a dynamic weighted scaling cross-entropy loss function in the literature [59,60]. Based on the binary cross-entropy loss corresponding to the object confidence loss and classification loss, a dynamic scaling factor can dynamically reduce the weights of the easily distinguishable samples during the training process through a dynamic scaling factor to quickly focus the weight of the loss on those difficult to differentiate samples, calculated as follows:(7)$$\begin{array}{c} WL=\left\{\begin{array}{cc} {\left(1-y^{{}^{\prime }}\right)}^{\eta }\left[-y\log y^{{}^{\prime }}-\left(1-y\right)\log\left(1-y^{{}^{\prime }}\right)\right], & y=1\\ y^{{}^{\prime }\eta }\left[-y\log y^{{}^{\prime }}-\left(1-y\right)\log\left(1-y^{{}^{\prime }}\right)\right], & y=0 \end{array}\right. \end{array}$$
where η is the attenuation coefficient, which is used to adjust the attenuation rate of simple samples. The default value is two, which can make the model pay more attention to those sample points that are harder to detect during the training process. The total loss after improvement is as follows:(8)$$\begin{array}{c} \begin{array}{cc} Loss & =\lambda_{coord}\sum_{i=0}^{S^{2}}\sum_{j=0}^{B}l_{ij}^{obj}\left[-t_{x_{i}}\log{\hat{t}}_{x_{i}}-\left(1-t_{x_{i}}\right)\log\left(1-{\hat{t}}_{x_{i}}\right)\right]\\ & +\lambda_{coord}\sum_{i=0}^{S^{2}}\sum_{j=0}^{B}l_{ij}^{obj}\left[-t_{y_{i}}\log{\hat{t}}_{y_{i}}-\left(1-t_{y_{i}}\right)\log\left(1-{\hat{t}}_{y_{i}}\right)\right]\\ & +\lambda_{coord}\sum_{i=0}^{S^{2}}\sum_{j=0}^{B}l_{ij}^{obj}\left[{\left(t_{w_{i}}-{\hat{t}}_{w_{i}}\right)}^{2}+{\left(t_{h_{i}}-{\hat{t}}_{h_{i}}\right)}^{2}\right]\\ & +\lambda_{obj}\sum_{i=0}^{S^{2}}\sum_{j=0}^{B}l_{ij}^{obj}\left[-c_{i}\log{\hat{c}}_{i}-\left(1-c_{i}\right)\log\left(1-{\hat{c}}_{i}\right)\right]\\ & +\lambda_{noobj}\sum_{i=0}^{S^{2}}\sum_{j=0}^{B}l_{ij}^{noobj}\left[-c_{i}{\left(1-{\hat{c}}_{i}\right)}^{\eta }\log{\hat{c}}_{i}-\left(1-c_{i}\right){\hat{c}}_{i}^{\eta }\log\left(1-{\hat{c}}_{i}\right)\right]\\ & +\lambda_{class}\sum_{i=0}^{S^{2}}l_{i}^{obj}\sum_{c\in class}\left[-p_{i}\left(c\right)\log{\hat{p}}_{i}\left(c\right)-\left(1-p_{i}\left(c\right)\right)\log\left(1-{\hat{p}}_{i}\left(c\right)\right)\right] \end{array} \end{array}$$
where $S^{2}$ is the total number of grid cells in the output feature map, *B* is the total number of predicted bounding boxes in each grid; $l_{ij}^{obj}$ and $l_{ij}^{noobj}$ are used to determine whether the *j* bounding box in the *i* grid contains the predicted object; $t_{x_{i}},t_{y_{i}},t_{w_{i}},t_{h_{i}}$ is the relative position parameter of the true bounding box, and ${\hat{t}}_{x_{i}},{\hat{t}}_{y_{i}},{\hat{t}}_{w_{i}},{\hat{t}}_{h_{i}}$ is the relative position parameter of the predicted bounding box; $c_{i}$ is the confidence level of the real bounding box, ${\hat{c}}_{i}$ is the confidence level of the predicted bounding box; $p_{i}\left(c\right)$ is the category probability of the real bounding box, ${\hat{p}}_{i}\left(c\right)$ is the category probability of the predicted bounding box; $\lambda_{coord}$ is the weight of coordinate loss in the total loss, the default value is 5; $\lambda_{obj}$ is the weight of positive samples in the confidence loss, and the default value is 1; $\lambda_{noobj}$ is the weight of negative samples in the confidence loss, and the default value is 0.5; $\lambda_{class}$ is the weight of category loss in the total loss, and the default value is 1.

## 4. Experiments

### 4.1. Experimental Environment

All experiments in this paper were conducted by Pycharm connected to a remote server with Ubuntu 18.04.3, a 12-core Intel (R) Xeon (R) Platinum 8255C 2.50GHz CPU, an NVIDIA Tesla V100 GPU, and 32GB of memory. Python 3.8, CUDA 11.0, PyTorch 1.8.1, PaddlePaddle 2.2.2, OpenMMLab’s MMDetection v2.25.2 [61], and PaddlePaddle’s PaddleDetection v2.5.0 [62].

### 4.2. Experimental Data and Preprocessing

Since there is no publicly available detection dataset for the fire and smoke object detection task, the experimental data in this paper are mainly derived from natural fire scenes and web video screenshots, including hundreds of real-life scenarios in total. Similar negative samples are considered to be added to the dataset to prevent the adverse effects caused by environmental factors, such as sun, clouds, smoke, lighting illumination, and others. The final experimental dataset of 14,904 photos was collected, with 13,843 images in the training set and 1061 images in the validation set, including a total of 116,709 fires and smoke target objects, some of which are displayed in Figure 7. To extract the coordinate information of the target items, we utilized the LabelImg program to label the dataset with the position coordinates, height and breadth, and target area. LabelImg merely labels the dataset in VOC format and transforms it to YOLO and COCO formats to fulfil the training needs of different models. To prevent model overfitting during training, the data enhancement methods include random rotation, random scaling, random cropping, random fusing, and change of saturation and chromaticity.

### 4.3. Parameter Settings

The model training process optimizer is the stochastic gradient descent (SGD) algorithm. The number of model training iterations is 300, the learning rate attenuation method is cosine annealing, and the batch size of the input model is 32. The model is not stable, thus warm-up is selected to preheat the learning rate. The specific parameters selected during the model training are provided in Table 1.

### 4.4. Experimental Results and Analysis

To ensure the interpretability of results across multiple models, we utilized $AP^{val}$ as the evaluation metric for the MS COCO dataset during our experiments. The models we compared include one-stage object detection algorithms, such as SSD, RetinaNet, FCOS, ATSS, YOLO, and PPYOLO, as well as two-stage object detection algorithms, such as Faster RCNN, Mask RCNN, Cascade RCNN, CenterNet, and DetectoRS. We also considered end-to-end object detection methods based on transformer structures, such as Detr, Deformable Detr, the ViTDetection family, and the SwinDetection family, as well as lightweight object detection algorithms such as PicoDet. For comparative testing, we followed the naming convention “Model backbone and neck”. All models were trained under the same experimental environment, image pre-processing methods, and hyperparameter settings. We initialized each model with pre-trained weights from the MS COCO dataset and taught it until convergence to accelerate the process and achieve the desired accuracy. The results are presented in Table 2.

After analysing the results in Table 2, it is evident that the one-stage object detection algorithm is considerably faster and has fewer parameters and computational complexity than the two-stage object detection algorithm. This makes it well-suited for meeting the requirements of real-time fire and smoke detection. However, the transformer-based object detection algorithm is complex and requires a large amount of training data to produce desirable results, which is limited by the data available in this study. In terms of detection accuracy and speed, YOLOR has the highest detection accuracy with an $AP^{val}$ of 61.3% and an FPS of 83.3; PicoDet has the fastest detection speed with an $AP^{val}$ of 47.9% and an FPS of 223.0. Compared to YOLOR, our model improves $AP^{val}$ by 1.2% and speeds up detection by 30. Compared to PicoDet, $AP^{val}$ improves by 14.6% based on the decrease in detection speed. Furthermore, compared with the original YOLOv7 structure, both detection accuracy and speed are improved. In addition, the $AP_{0.5}^{val}$ in Table 2 provided that for small objects in the fire smoke dataset that take up less than 32 × 32 pixels, the highest accuracy of 58.5% of all models was achieved, demonstrating the accuracy of the model for fire source detection. Although it is slightly inferior in detection speed compared to some fast detection models, it still meets the real-time fire smoke detection criteria. Compared with the other models in the table, the PSA module focuses on extracting features from fire and smoke textures and can accurately capture remote contextual information in both horizontal and vertical directions, improving the effect of feature extraction. For the variable shape, characteristics, and scale of fire and smoke objects, the RC FPN-PAN allows the predicted bounding box to better fit the true bounding box during detection, thus improving the $AP^{val}$.

Considering the specificity of fire smoke detection, which requires both accuracy and speed, a comparison of the speed and accuracy of different variants of the one-stage detection algorithm was thus performed, as shown in Figure 8, where our model has a 62.5% map and 113.6 FPS on V100, placing it ahead of other one-stage detection methods. To further validate the model’s accuracy, the results of standard one-stage detection methods were evaluated using test time augmentation (TTA), a technique used to improve the performance of neural network models and reduce generalization errors [63]. The training set was extended by using modified copies of the samples from the training dataset, the original data was augmented in different forms, and then the average of each result was taken as the final result, as provided in Table 3. On the fire smoke dataset, the highest accuracy our model could achieve after using TTA was 65.1%, 1.2% higher than the accuracy of YOLOR-CSP-X after using TTA.

To investigate the effect of different parameter setting thresholds on the detection accuracy of the model, we try to change the model’s accuracy training method and IoU threshold. The default training accuracy is FP16 with a half-accuracy training method, and the default IoU threshold is 0.65. The results are provided in Table 4. When the accuracy training method and IoU threshold are changed, the model detection accuracy is slightly improved, and the $AP^{val}$ reached 62.7%, 0.2% higher than the default setting parameters.

In real-world applications, it is essential to consider the detection accuracy of a detector for a single category of objects. The fire and smoke detection results are presented in Table 5. YOLOv7-X demonstrates the highest fire detection accuracy among the one-stage detectors, achieving 62.0%. For smoke detection, YOLOR-CSP-X achieves the highest accuracy rate at 60.6%. Among the two-stage detectors, Cascade-Mask-ConvNeXt has the highest accuracy rates of 58.8% and 55.4% for fire and smoke detection, respectively. In the transformer-based detectors, Mask-RepPoints-Swin achieves the highest detection accuracy rates of 55.8% and 51.8% for fire and smoke detection, respectively. Our improved model surpasses the best results, achieving detection accuracy rates of 63.9% and 61.1% for fire and smoke, respectively, which are 1.9% and 0.5% higher than the previous best results.

To assess the generalization performance of the proposed structure, all variants of YOLOv7 were trained using an enhanced method. The results obtained are presented in Table 6, with a “+” signifying an improved model. Our enhanced approach achieves a higher detection accuracy while reducing the model’s complexity and the number of parameters. The top accuracy attained by $AP^{val}$ was 62.5% and 64.3% for input image sizes of 640 × 640 and 1280 × 1280, respectively.

To highlight the regions of focus in the image, the weights corresponding to the class feature maps are determined by intercepting the gradient information during the backpropagation of the model. These weights are then superimposed onto the original map along with the feature maps. This technique, called gradient-weighted class activation mapping (Grad-CAM), is utilized to visualize the model [64]. The effectiveness of our model in accurately locating the object of interest in the image can be observed in Figure 9, demonstrating superior performance compared to other one-stage object detection algorithms.

### 4.5. Ablation Study

The three-part interactions, CDPB, PSA, and RC FPN-PAN, were considered to improve the model’s accuracy. The experiments were conducted under the same experimental environment, image pre-processing, and network hyperparameter settings and the results are provided in Table 7. The $AP^{val}$ changes to 0.3%, 1.1%, and 1.7% when acting on CDPB, PSA, and RC FPN-PAN, respectively. The highest accuracy is reached when acting together with an $AP^{val}$ of 62.3%, which is a 2.9% improvement compared to the improved original model.

This paper uses two model improvement strategies: the sample matching strategy across the grid and the loss function with weighted decay. The ablation experiment results are provided in Table 8 to verify their effectiveness. When the two strategies were used, the $AP^{val}$ was improved by 1.2% and 1.8%, respectively, compared to the original model improvement of the $AP^{val}$ by 2.6%.

Table 9 presents the results of our evaluation on the impact of the improved structure and strategy of the model on its detection accuracy. Compared to the original model’s improved $AP^{val}$, the improved structure and strategy resulted in a 2.9% and 2.6% increase, respectively. They produced a 3.1% improvement in $AP^{val}$ when used together.

The results of our ablation experiments provide further evidence that our improved method enhances the accuracy of fire smoke detection and effectively addresses the issue of scale change in flames and smoke. To illustrate the performance of our one-stage object detector, we present a visualization of the results on the validation set in Figure 10. The “Pictures” in the figure represent the ground-truth bounding boxes corresponding to the original image. Upon comparison of each detector’s output with the original image, we observe that YOLOv5, YOLOv6, and YOLOR all missed the detection of the flame object in the lower left corner of the first image. YOLOR incorrectly detects the light solid area in the second image as a flame object. Moreover, YOLOv5 and YOLOv7 exhibit poor detection performance on the overlapping regions of the second and fourth images. In the original image of the fourth example, there are five target objects, yet both YOLOv5 and YOLOv7 detect only four objects.

## 5. Conclusions and Outlook

Deep learning-based fire smoke detection algorithms have become increasingly crucial for practical applications in recent years. This paper proposes an improved algorithm based on the YOLOv7 model, which combines several features to enhance accuracy and speed, such as an attention mechanism, multi-scale feature fusion and prediction, a sample matching strategy, and a loss function with weight attenuation. A new fire smoke dataset was created using natural fire scenes and web video screenshots with various positive and negative samples to develop this algorithm. The structure of the model was improved to increase detection efficiency using the CDPB structure to reduce the number of model parameters and speed up reasoning, the PSA structure to enhance information fusion between channels and improve generalization performance, and the RC FPN-PAN structure to address scale changes in flame and smoke objects. In addition, the cross-grid sample matching strategy and the weighted attenuation loss function were redesigned to improve the prediction bounding box’s accuracy and accelerate model convergence. The experiment used a training set with 13,843 images and 112,576 fire smoke detection objects and a validation set with 1061 images and 4133 fire smoke detection objects. The detection result of the improved model has an $AP^{val}$ of 62.5% and an FPS of 113.6, outperforming other methods. Notably, most high-accuracy models achieved on public datasets for object detection, such as MS COCO, are based on Transformers that require a large amount of data for training to achieve desired results. However, this may not be feasible for our fire smoke dataset.

Our future work will focus on collecting more fire and smoke images in complex environments to uncover additional correlations between the environment’s morphological and detailed features and the characteristics of the smoke. We plan to combine semantic information from the environment with its characteristic information to improve the detection accuracy. Furthermore, we will consider the colour features of smoke, such as white, black, or yellow, to make rough judgments about the potential harmfulness of the fire. For instance, if the smoke is white, it may indicate a high dust content; black smoke suggests incomplete combustion and a large amount of sulphide in the flue gas; yellow smoke can also signify high sulphide content, and the colour is generally indicative of highly toxic and corrosive substances.

## Figures and Tables

**Figure 1 sensors-23-03358-f001:**
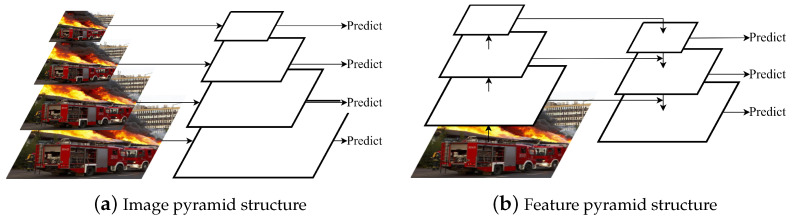
Two types of pyramid structures commonly used in tasks involving the detection of objects in realistic scenes.

**Figure 2 sensors-23-03358-f002:**
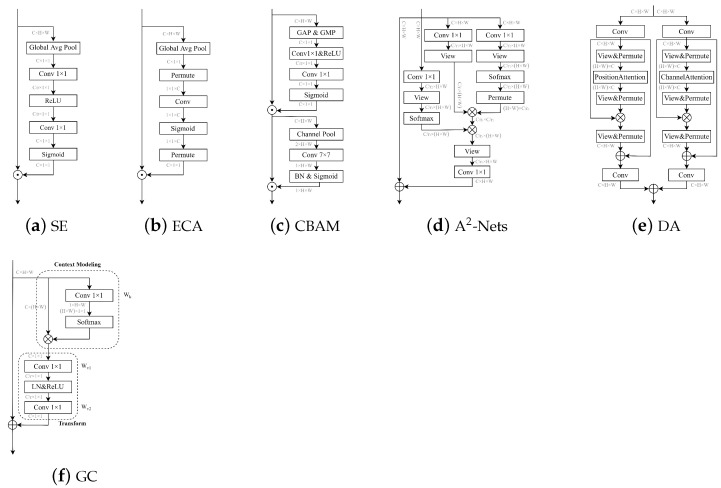
This diagram illustrates the framework of a model for three main forms of soft attention processes. Sub-figures (**a**,**b**) represent channel attention, (**c**,**d**) represent spatial attention, while (**e**,**f**) represent self-attention.

**Figure 3 sensors-23-03358-f003:**
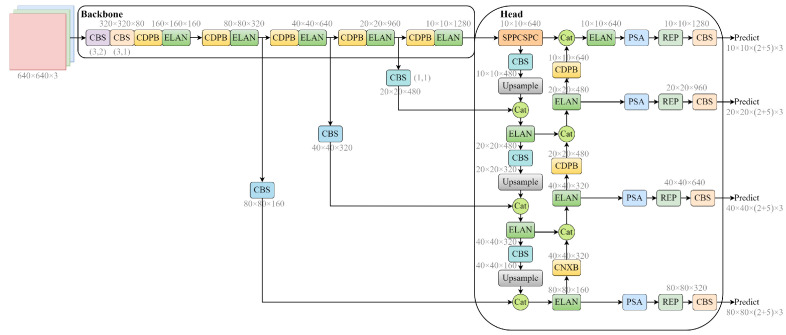
The structure of the improved fire smoke detection model based on YOLOv7-X.

**Figure 4 sensors-23-03358-f004:**
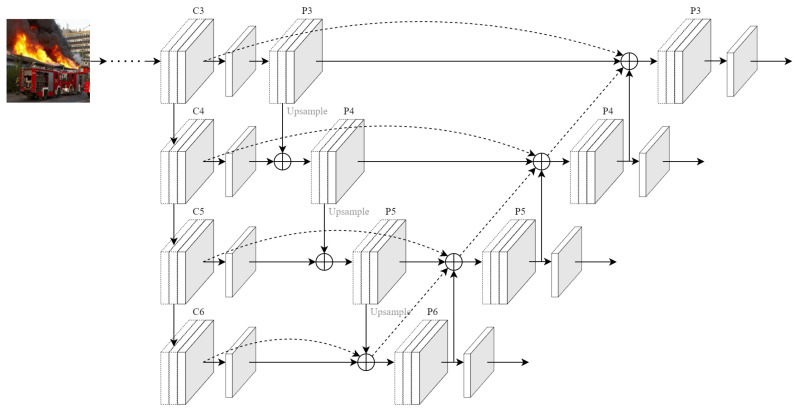
Radially connected FPN-PAN structure (RC FPN-PAN) operating by stitching the feature information from the initial layer to the final prediction layer through the residual structure of the dashed line.

**Figure 5 sensors-23-03358-f005:**
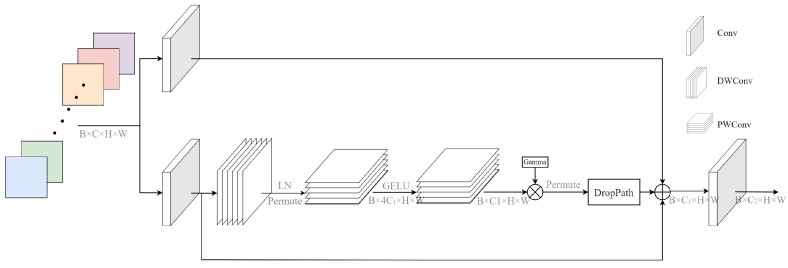
Diagram illustrating the CDPB module’s structure, utilizing DWConv and PWConv structures in place of the original convolutional layers (We have chosen kernel_size equal to three, and the CDPB is about one-ninth the computational effort of full convolution).

**Figure 6 sensors-23-03358-f006:**
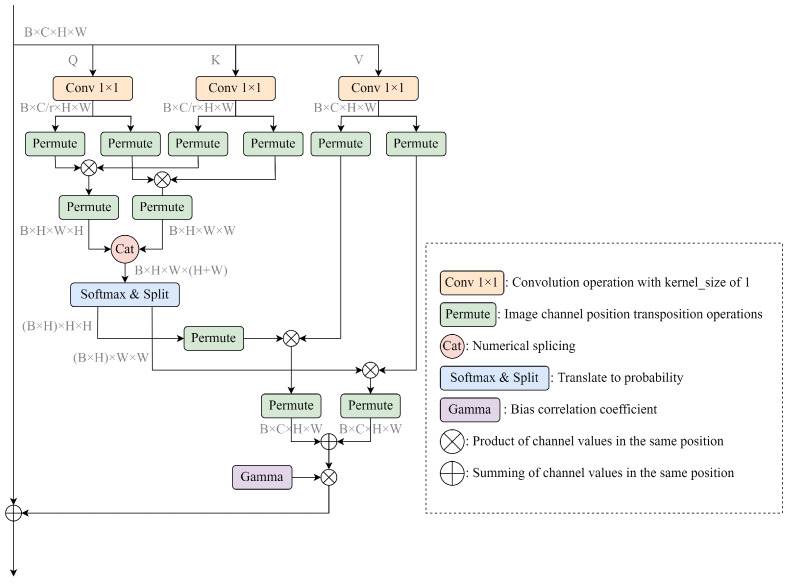
A permutation self-attentive mechanism is utilized to process both channel and spatial feature information.

**Figure 7 sensors-23-03358-f007:**
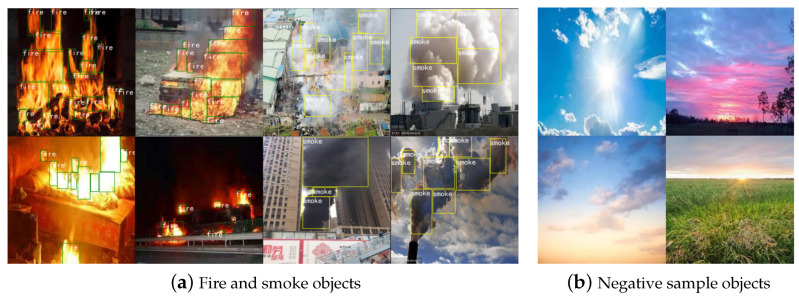
Schematic diagram of a partial dataset, with (**a**) real target objects to be detected and (**b**) negative samples that are susceptible to interference.

**Figure 8 sensors-23-03358-f008:**
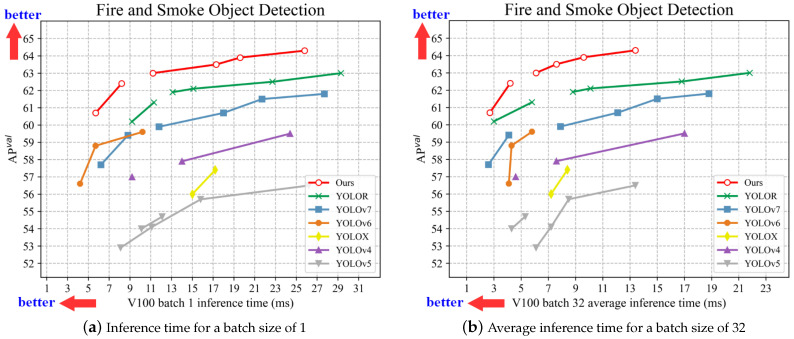
Comparison of results with one-stage object detection methods in terms of inference speed.

**Figure 9 sensors-23-03358-f009:**
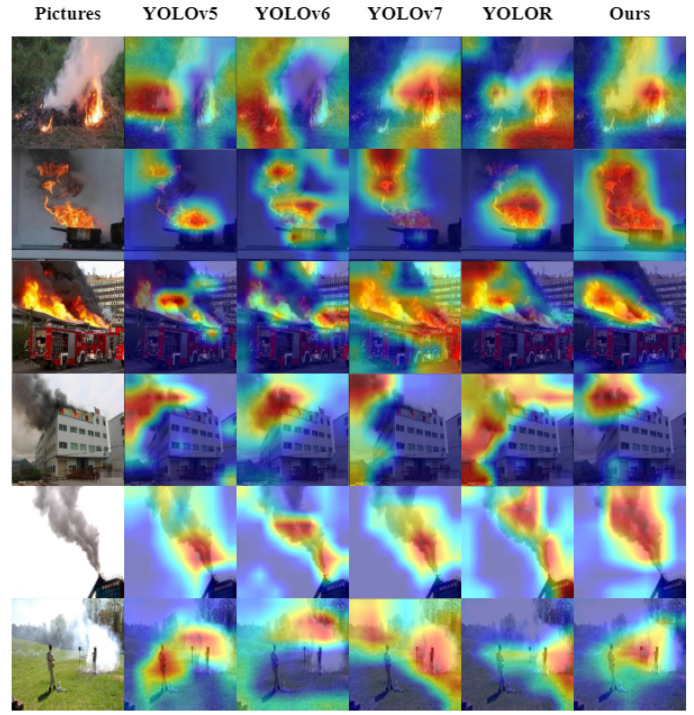
Comparison of heatmaps results with one-stage object detectors. The deeper the colour of the area the stronger the attention.

**Figure 10 sensors-23-03358-f010:**
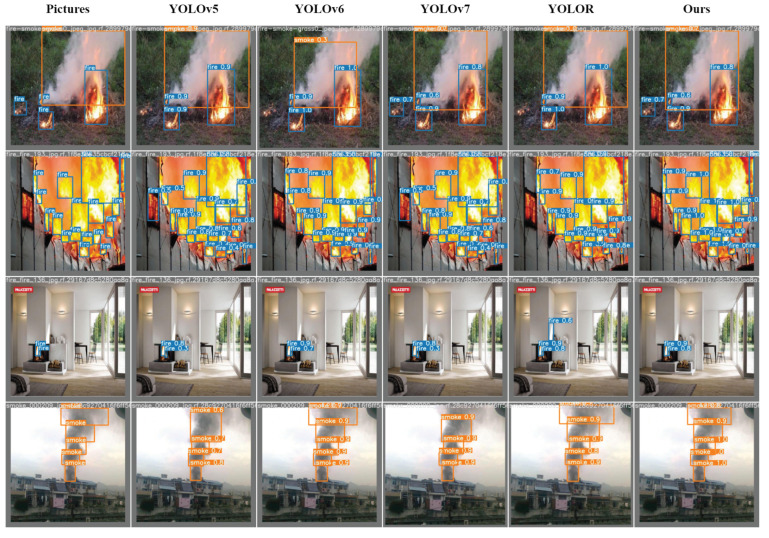
Comparison of results with standard one-stage object detectors.

**Table 1 sensors-23-03358-t001:** The main parameters of the model training process and their corresponding values.

Parameter Name	Parameter Value
epoch	300
batch size	32
learning rate	0.001
weight decay	0.0005
optimizer	SGD
momentum	0.937
warmup	3
random seed	42
image size	640
num workers	12

**Table 2 sensors-23-03358-t002:** Comparison results of our constructed model with standard one-stage object detection methods, two-stage object detection methods, and transformer-based object detection methods on the constructed dataset.

Method	Year	${\mathrm{AP}}^{\mathrm{val}}$	${\mathrm{AP}}_{0.5}^{\mathrm{val}}$	${\mathrm{AP}}_{0.75}^{\mathrm{val}}$	${\mathrm{AP}}_{S}^{\mathrm{val}}$	${\mathrm{AP}}_{M}^{\mathrm{val}}$	${\mathrm{AP}}_{L}^{\mathrm{val}}$	${\mathrm{AR}}^{\mathrm{val}}$	# Param. (M)	FLOPs (G)	FPS (V100)
**One-Stage Detectors**											
SSD	2016	30.8	65.9	24.6	10.6	25.6	40.8	43.5	24.5	87.9	24.5
PPYOLO-R50	2020	33.8	68.6	29.2	28.2	35.0	37.3	54.3	46.6	106.6	155.6
RetinaNet-R101-FPN	2017	35.1	71.2	31.6	6.9	36.5	41.6	44.9	55.1	112.3	175.9
YOLOv3-SPP	2018	35.1	76.8	28.1	33.1	36.2	38.0	44.8	62.0	78.2	200.6
FCOS-R101-FPN	2019	37.7	70.1	37,3	19.3	40.5	42.1	48.4	50.8	109.1	171.1
RetinaNet-FreeAnchor	2017	40.4	75.7	39.7	36.7	40.1	44.9	49.4	56.7	126.1	177.3
ATSS-R101-FPN	2020	41.8	71.7	45.0	32.4	43.0	45.9	51.5	50.9	111.0	172.1
VarifocalNet-R101-FPN	2021	42.1	73.1	44.3	35.5	43.0	47.1	51.5	51.5	106.0	164.7
PPYOLOE-X	2022	42.9	76.1	44.4	38.7	45.0	47.4	58.6	95.3	204.9	95.2
PPYOLOv2-R101	2021	44.5	78.8	46.2	53.4	47.8	46.4	61.8	73.2	187.4	87.0
PicoDet-L	2021	47.9	82.1	50.2	39.9	49.2	51.3	58.0	**5.8**	**16.8**	223.0
RTMDet-X	2022	48.1	84.2	50.7	27.3	45.1	54.3	57.3	94.9	141.7	136.1
PPYOLOE+-X	2022	49.6	82.3	54.3	37.1	49.8	54.7	62.6	98.4	206.6	95.2
TOOD-R101-FPN	2021	52.1	82.3	57.8	36.3	53.9	56.2	60.3	53.4	73.3	175.5
ATSS-FPN-DyHead	2019	52.4	84.3	58.1	28.7	52.5	58.5	61.4	210.4	322.2	50.1
YOLOv5-L6	2020	56.5	89.0	61.4	38.4	52.6	61.9	64.3	76.2	110.5	79.4
YOLOv4-P5	2020	57.0	90.0	65.5	46.0	55.9	64.7	67.2	70.3	166.0	105.3
YOLOX-X	2021	57.4	86.8	65.8	55.6	56.1	60.5	63.3	99.0	282.0	38.6
YOLOv7-X	2022	59.4	89.3	67.8	56.1	56.8	64.7	66.8	70.8	188.9	108.7
YOLOv6-L	2022	59.6	88.8	68.1	49.4	57.8	64.8	67.5	58.5	143.8	94.3
YOLOR-CSP-X	2021	61.3	**91.0**	70.2	49.9	58.1	66.6	**68.4**	99.8	223.0	83.3
**Two-Stage Detectors**											
Faster-R50-PAFPN	2018	29.1	63.0	22.1	19.0	28.9	34.3	37.5	44.7	100.9	175.7
Sparse-R101-FPN	2020	34.8	69.7	26.3	20.5	33.4	41.1	60.8	124.9	95.1	174.0
CenterNet-R18-DCN	2019	34.8	71.3	30.1	15.9	34.8	39.7	46.7	14.4	19.3	71.3
Mask-R101-FPN	2017	35.7	69.0	33.4	35.7	37.7	40.4	43.3	63.2	174.9	13.5
Faster-R101-FPN	2017	37.2	73.8	32.5	27.0	38.3	42.5	44.7	60.1	121.4	177.2
DDOD-R50-FPN	2021	37.8	69.8	37.6	27.5	37.4	43.2	47.0	32.2	72.7	172.3
Cascade-R101-FPN	2018	38.5	72.2	36.7	20.8	38.3	45.0	45.6	87.9	149.2	176.0
Cascade-X101-DCN-FPN	2019	42.0	73.2	43.8	19.5	43.1	48.7	49.0	95.4	305.2	177.6
RepPoints-R101-FPN	2019	45.3	77.6	47.6	22.5	45.0	52.6	53.6	56.9	81.2	170.7
Libra Faster-X101-FPN	2019	47.2	75.7	53.4	30.6	47.8	52.1	54.1	99.1	184.8	178.5
DetectoRS-HTC-R101	2020	47.8	78.9	52.1	25.5	46.1	54.5	56.1	196.5	279.1	49.6
HTC-Res2Net-R101-FPN	2019	48.9	78.5	54.6	33.8	48.4	54.6	55.3	89.2	156.7	4.4
RetinaNet-R50-FPG	2020	48.9	81.7	53.4	16.3	51.4	55.1	57.4	70.9	122.2	205.9
Cascade-Mask-ConvNeXt-S	2021	49.7	82.3	50.2	30.1	53.2	55.7	57.7	120.4	133.3	78.5
Mask-ConvNeXt-T	2021	51.1	83.8	56.4	36.6	54.1	56.2	57.9	99.2	150.8	89.5
Faster-R50-FPG	2020	53.7	85.0	60.4	45.2	54.4	57.2	59.5	79.4	253.7	214.5
Mask-R50-FPG	2020	55.2	85.4	62.9	48.7	55.4	59.3	60.6	82.0	305.2	44.7
Cascacde-Mask-S101-FPN	2020	56.1	86.4	64.6	48.5	57.6	62.1	62.4	103.2	211.5	56.8
Cascade-Mask-ConvNeXt-T	2021	57.6	86.8	66.1	49.8	57.9	63.5	64.3	124.7	234.8	48.9
**Transformer-Based Detectors**											
Detr-R50	2020	35.6	72.3	30.7	20.9	32.8	43.0	50.7	41.3	37.1	55.6
Deformable-Detr-R50	2020	37.8	75.1	36.7	23.7	33.5	45.6	55.4	55.8	45.9	57.4
ViTDet-Base	2020	47.8	76.5	47.3	34.5	41.2	50.1	57.8	106.3	210.5	66.2
Mask-R50-Swin-S	2021	48.9	77.7	49.6	37.9	43.1	51.7	58.8	67.5	230.6	68.9
ViTDet-Faster-R50	2020	50.2	80.3	52.7	37.4	45.8	53.8	59.2	77.4	284.7	74.8
RepPoints-Swin-T	2021	52.1	82.6	54.0	41.3	46.8	56.1	62.3	88.5	189.5	77.4
ViTDet-Large	2020	53.3	83.2	55.7	42.0	48.8	58.9	62.4	156.7	238.5	44.0
Mask-RepPoints-Swin-T	2021	53.8	84.7	57.4	45.1	52.1	57.5	64.3	101.6	200.3	68.4
**Ours**		**62.5**	87.9	**71.4**	**58.5**	**60.7**	**68.1**	68.1	45.4	143.6	113.6

**Table 3 sensors-23-03358-t003:** Results of the one-stage object detection methods when TTA is used or not (✓ denotes use, − denotes not).

Method	TTA	Size	${\mathrm{AP}}^{\mathrm{val}}$
YOLOv5-L6	−	640	56.5
	✓	640	58.7
YOLOv4-P5	−	640	57.0
	✓	640	58.6
YOLOv7-X	−	640	59.4
	✓	640	59.7
YOLOv6-L	−	640	59.6
	✓	640	60.3
YOLOR-CSP-X	−	640	61.3
	✓	640	63.9
Ours	−	640	**62.5**
	✓	640	**65.1**

**Table 4 sensors-23-03358-t004:** Results of different precision and IoU threshold.

Method	Precision	IoU Threshould	${\mathrm{AP}}^{\mathrm{val}}$
Ours	FP16 (default)	0.65 (default)	62.5
	FP32	0.65	62.5
	FP16	0.70	62.5
	FP32	0.70	62.7
Improvement			**+0.2**

**Table 5 sensors-23-03358-t005:** Fire and smoke object detection results.

Method	${\mathrm{AP}}^{\mathrm{val}}$	Fire	Smoke
**One-Stage Detectors**			
YOLOv5-L5	56.5	56.1	56.9
YOLOv4-P5	57.0	57.3	56.7
YOLOX-X	57.4	61.9	53.0
YOLOv7-X	59.4	62.0	56.7
YOLOv6-L	59.6	60.6	58.7
YOLOR-CSP-X	61.3	61.9	60.6
**Two-Stage Detectors**			
Faster-R50-FPG	53.7	56.1	51.2
Mask-R50-FPG	55.2	57.5	53.0
Cascade-Mask-S101-FPN	56.1	58.1	54.1
Cascade-Mask-ConvNeXt-T	57.6	58.8	55.4
**Transformer-Based Detectors**			
RepPoints-Swin-T	52.1	55.7	48.5
ViTDet-Large	53.3	55.4	51.2
Mask-RepPoints-Swin-T	53.8	55.8	51.8
Ours	**62.5**	**63.9**	**61.1**

**Table 6 sensors-23-03358-t006:** Improved results for different detection scales for YOLOv7 variants.

Method	Size	${\mathrm{AP}}^{\mathrm{val}}$	#Param.(M)	FLOPs(G)
YOLOv7	640	57.7	36.9	104.7
YOLOv7+	640	60.7	21.6	35.7
YOLOv7-X	640	59.4	71.3	189.9
YOLOv7-X+	640	62.5	37.1	70.2
YOLOv7-W6	1280	59.9	81.0	360.0
YOLOv7-W6+	1280	63.0	39.9	100.3
YOLOv7-E6	1280	60.7	97.2	515.2
YOLOv7-E6+	1280	63.5	45.4	144.5
YOLOv7-D6	1280	61.5	154.7	806.8
YOLOv7-D6+	1280	63.9	78.8	187.9
YOLOv7-E6E	1280	61.8	151.7	843.2
YOLOv7-E6E+	1280	64.3	77.7	206.8

**Table 7 sensors-23-03358-t007:** Results of ablation experiments with improved structures.

Rank	CDPB	PSA	RC FPN-PAN	${\mathrm{AP}}^{\mathrm{val}}$
1	−	−	−	59.4
2	✓	−	−	59.7
3	−	✓	−	60.5
4	−	−	✓	61.1
5	✓	✓	−	61.7
6	−	✓	✓	62.0
7	✓	−	✓	62.1
8	✓	✓	✓	62.3
Improvement				**+2.9**

**Table 8 sensors-23-03358-t008:** Results of ablation experiments with improved strategies.

Rank	Match Strategy	Weight Decay	${\mathrm{AP}}^{\mathrm{val}}$
1	−	−	59.4
2	−	✓	60.6
3	✓	−	61.2
4	✓	✓	62.0
Improvement			**+2.6**

**Table 9 sensors-23-03358-t009:** Results of ablation experiments with improved structures and strategies.

Rank	Structure	Strategy	${\mathrm{AP}}^{\mathrm{val}}$
1	−	−	59.4
2	−	✓	62.0
3	✓	−	62.3
4	✓	✓	62.5
Improvement			**+3.1**

## Data Availability

Experimental data covered in this paper, including improved model structures, comparison models, and datasets, are available at the following https://github.com/jinc1216/fire-smoke-detection accessed on 1 March 2023; for validation codes, please contact the authors by email. Three different dataset formats are included: VOC, YOLO, and COCO.

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
