# Peer review of "Real-Time Fire Smoke Detection Method Combining a Self-Attention Mechanism and Radial Multi-Scale Feature Connection"

_sensors, 2023, doi:10.3390/s23063358_

Round 1

Reviewer 1 Report

In this manuscript entitled “Real-time Fire Smoke Detection Method Combine Self-attention

Mechanism and Radial Multi-scale Feature Connection” (Manuscript Number: sensors-2261488) I think it’s better to discuss about below questions. Therefore, I suggest a major revision for the manuscript before publication.

Comments:

  1. All equations should be referred in the text using a number.
  2. The grammatical errors should be corrected. For example "by" or "using" instead "by using" and etc.
  3. The English language should be improved.
  4. The authors should be explaining about the importance and novelty of the work with more details.
  5. A comparison table should be presented to show the novelty of the manuscript.

Reviewer 2 Report

This paper proposes an improved algorithm based on the YOLOv7 model, which combines several features to enhance accuracy and speed, such as an attention mechanism, multi-scale feature fusion and prediction, a sample matching strategy, and a loss function with weight attenuation.  

Further ablation studies are performed to optimise the network for optimal results and conclusions are drawn. The results are well compared with other methods on a wide variety of data sets. The results obtained are well presented and perform better than most previous methods. However, a minor revision is needed before acceptance. Please see my comments below:

It would be more comprehensive to illustrate in more detail how the module, such as the CDPB, reduces the number of model parameters and speeds up the reasoning process.

Reviewer 3 Report

This paper proposes a novel method aiming at the real-time fire smoke detection issue basing on the YOLOv7 detector. Firstly, it proposes the radial connection on the basics of FPN and PAN to enhance the semantic and location information of the feature. Then it adopts permutation self-attention mechanism to gather contextual information as accurately as possible. Besides, it proposes the CDPB module to make a better extraction for the feature. Finally, the improved loss function and sample method are proposed to make a better detection.

Comments on Review:

(1)   It’s better to polish the abstract, and briefly summary the issue of fire and smoke.

(2)   The “targe object” in the third paragraph of the introduction is repeated and there is a grammatical error of the “have proven”.

(3)   The full name of proposed module CDPB in this paper is not mentioned.

(4)   The “Warmup” in the 4.3. Parameter Settings is not suitable, may be the “warmup”.

(5)   The compared methods used for visualizing are all the one stage based, how about two stages?

(6)   It’s better to explain the issue of failure to detect for fire and smoke with figures.

(7)   At the end of introduction, it's better to mention all the other methods proposed in this paper not only the feature fusion.

(8)   The authors talk too much about the existing methods, whereas the improvement of the algorithm needs to be highlighted.

(9)   If possible, it’s better to conduct the experiment on another fire and smoke dataset to verify the generalization of proposed method.

Reviewer 4 Report

Modern theory and algorisms of machine learning are implemented for fire smoke detection. Necessary modifications for fire detection are provide.

The reviewer hope to get answer to the following questions:

When does an initially small fire source become a full-blown fire?

What are differences among: fire smoke detection, fire detection, smoke detection?

What does ‘ablation’ mean in ‘Ablation Study’?

Why is the ‘gradient-weighted’ method work better?

(typing error) In table 2: ‘RetenaNet’ -> ‘RetinaNet’

Round 2

Reviewer 3 Report

The authors have addressed my concerns. Now this paper can be published.

Reviewer 4 Report

This reviewer raised four items of concerns about the initial submission.

The authors made a detailed and amusing set of answers thereto.

The authors seem to have a solid grasp of their study and contents.